# When Fungal Prophylaxis Fails: A Rare Case of *Rhodotorula mucilaginosa* Fungemia with Suspected Abdominal Origin

**DOI:** 10.3390/jof11100723

**Published:** 2025-10-08

**Authors:** Elia Asensi-Díaz, Laura Barbero del Olmo, Patricia Urrutia, Ana Lario, Elia Gómez-G. de la Pedrosa, Alejandro G. García-Ruiz de Morales, Pilar Martín-Dávila, Jesús Fortún

**Affiliations:** 1Infectious Diseases Division, Fundación Jiménez Díaz University Hospital, 28040 Madrid, Spain; 2Haematology Department, Ramón y Cajal Hospital, 28034 Madrid, Spain; 3Infectious Diseases Department, Ramón y Cajal Hospital, 28034 Madrid, Spain; 4Torrecárdenas University Hospital, 04009 Almeria, Spain; 5Instituto de Investigación Sanitaria Hospital “Ramón y Cajal” (IRYCIS), 28034 Madrid, Spain; 6Microbiology Department, Ramón y Cajal Hospital, 28034 Madrid, Spain; 7Centro de Investigación Biomédica en Red de Enfermedades Infecciosas (CIBERINFEC), Instituto de Salud Carlos III, 28029 Madrid, Spain; 8Department of Medicine, Universidad Alcalá, 28801 Madrid, Spain

**Keywords:** *Rhodotorula*, fungemia, neutropenia, antifungal prophylaxis

## Abstract

We report a rare case of *Rhodotorula mucilaginosa* fungemia with a suspected abdominal origin in a 73-year-old man with advanced haematological disease on fluconazole prophylaxis. The patient presented with febrile neutropenia caused by a jejunal microperforation. Despite broad-spectrum antibiotics, the fever persisted, and *Rhodotorula mucilaginosa* was isolated from blood cultures. High-dose liposomal amphotericin B achieved microbiological clearance and clinical improvement. The case was further complicated by coinfection with *Aspergillus fumigatus* and *Klebsiella oxytoca*. To our knowledge, this is one of the few reported cases of abdominal *Rhodotorula* fungemia, and the first described in the context of fluconazole prophylaxis. This report emphasises the importance of recognising *Rhodotorula* as a true pathogen and highlights the challenges of managing rare fungal infections in immunocompromised hosts.

## 1. Introduction

*Rhodotorula* species are basidiomycetous yeasts commonly found in the environment and classified within the Pucciniomycotina subphylum [1]. For many years, they were considered harmless colonisers or contaminants, but clinical evidence has demonstrated that they can act as opportunistic pathogens, particularly in immunocompromised hosts [2,3,4]. The most frequent presentation is fungemia, although cases of peritonitis, meningitis, and endocarditis have also been documented [4]. We report a rare case of *Rhodotorula mucilaginosa* fungemia with a suspected abdominal origin in a neutropenic patient.

## 2. Case Description

We describe the case of a 73-year-old man with a history of chronic myelomonocytic leukaemia type 2, which progressed to acute myeloid leukaemia (AML) with a mast cell component. He had received avapritinib and azacitidine, with the last cycle administered 16 months before admission. Due to persistent neutropenia, he was receiving fluconazole 200 mg daily as antifungal prophylaxis. He was admitted for febrile neutropenia, caused by a contained jejunal microperforation identified during staging for advanced systemic mastocytosis via a body computed tomography (CT) scan.

The patient experienced daily afternoon fevers (>38 °C) and intermittent crampy abdominal pain in the days before admission. On physical examination, his abdomen was soft and tender throughout but showed no peritoneal irritation. No additional significant findings were noted, and he did not meet criteria for sepsis at this point. Laboratory results showed pancytopenia with grade IV neutropenia (haemoglobin 9.45 g/dL, platelets 113,000/μL, leukocytes 400/μL, neutrophils 20/μL), along with moderately elevated inflammatory markers (C-reactive protein 167 mg/L and procalcitonin 4.8 ng/mL). No other organ dysfunction was detected. The General Surgery team recommended conservative management with radiological follow-up. Empirical antibiotics—piperacillin-tazobactam 4/0.5 g every 6 h and amikacin (1 g single dose)—were initiated.

Due to persistent fever, the patient was reassessed. Vancomycin (1 g every 12 h) was initiated at this stage to cover Gram-positive pathogens, including MRSA, given the cellulitis and the patient’s risk factors related to severe immunosuppression and frequent hospital exposure. Blood cultures taken multiple times remained persistently negative. Urine culture, chest X-ray, and respiratory viral panel were all negative.

Despite nearly two weeks of broad-spectrum antibiotics, the patient continued to have daily fevers with chills, and no other source of infection was identified. At this point, an Infectious Diseases consultation was requested. On our evaluation, the patient was haemodynamically stable, with persistent abdominal pain and signs suggestive of peritoneal irritation on palpation. The cellulitis on the left forearm showed signs of improvement. The patient had a peripherally inserted central catheter (PICC) with two lumens in good condition. New blood cultures were obtained from both a peripheral vein and the PICC lines. Given the ongoing fever, severe neutropenia, and abdominal findings, antibiotic therapy was escalated to meropenem (2 g every 8 h), and empirical antifungal treatment with anidulafungin (200 mg loading dose, followed by 100 mg daily) was started.

The General Surgery team was consulted again. A follow-up abdominal CT showed stability of the jejunal microperforation, and conservative management was continued.

Although the patient remained in good general condition, he continued to have daily fevers. After five days, *Rhodotorula mucilaginosa* was isolated from 1 of 3 peripheral blood culture bottles. Antifungal therapy was escalated to liposomal amphotericin B (5 mg/kg every 24 h). Forty-eight hours later, while still receiving anidulafungin, *Rhodotorula* was isolated again—this time in 2 of 3 blood culture bottles. Susceptibility testing showed resistance to azoles and echinocandins, as expected for this genus, and a minimum inhibitory concentration (MIC) of 1 µg/mL for amphotericin B. In detail the isolate showed the following MICs: fluconazole >256 µg/mL, itraconazole 2 µg/mL, voriconazole 8 µg/mL, posaconazole 2 µg/mL, isavuconazole 1 µg/mL, and echinocandins >8 µg/mL.

The addition of 5-flucytosine or a compassionate-use request for fosmanogepix was considered. However, high-dose liposomal amphotericin B monotherapy was continued due to the patient’s clinical improvement and clearance of blood cultures after 48 h of treatment.

A transthoracic echocardiogram and fundoscopy revealed no evidence of endocarditis or ocular involvement. Chest CT showed no pulmonary consolidations. The PICC line was also replaced as part of the management strategy, and no microorganisms were isolated from the catheter tip culture.

Despite clinical and microbiological improvement, the patient continued to experience daily fever. On re-examination, the lesion on the right forearm worsened and developed into an ulcer (Figure 1). A skin biopsy was performed, and culture of the sample grew *Klebsiella oxytoca* (OXA-48 carbapenemase resistance pattern) and *Aspergillus fumigatus.* This was the only biopsy, and histopathological analysis confirmed true infection by both pathogens, ruling out contamination.

After two weeks of treatment with meropenem, antibiotic coverage was escalated to include ceftazidime–avibactam (2 g every 8 h), and isavuconazole was added (200 mg every 8 h for 48 h as a loading dose, then 200 mg daily). Isavuconazole was chosen over voriconazole because it offered equivalent efficacy with a more favourable safety profile, fewer drug interactions, and simpler management in this immunocompromised patient. Vancomycin had been discontinued after 10 days of treatment, and liposomal amphotericin B after 14 days from blood culture clearance, once the patient was in good general condition. The persistent fever was considered more likely related to his underlying haematological disease, which had not yet been treated due to the intercurrent infection. Susceptibility testing confirmed that Aspergillus was susceptible to amphotericin B.

Despite ongoing treatment, the patient continued to have a fever, so a Positron Emission Tomography–Computed Tomography (PET-CT) showed increased 18F-fluorodeoxyglucose (18F-FDG) uptake in the left upper lobe of the lung, in the right arm (at the site of the skin lesion), and improvement in the jejunum. This uptake in the lung is not specific and can also be seen with bacterial infection, inflammation, or malignancy. Bronchoscopy was considered as the next diagnostic step, but it was not performed because the patient had severe thrombocytopenia and was very frail, which made the procedure unsafe. Although galactomannan and beta-D-glucan tests were negative, the PET-CT findings, together with the skin biopsy positive for *Aspergillus fumigatus,* led us to treat the case as probable invasive aspergillosis.

After infection resolution, persistent daily fever was interpreted as possibly related to disease progression. A first cycle of azacitidine was initiated, which resulted in the resolution of the fever.

The patient initially showed clinical improvement, with decreasing inflammatory markers, and was discharged after seven days of ceftazidime–avibactam and isavuconazole therapy. Isavuconazole was continued as outpatient antifungal treatment with close follow-up by the Infectious Diseases team. One week later, he was reassessed by the Haematology department and presented with marked deterioration, having lost independence in basic activities of daily living and any ability to communicate. During this period, he reported a low-grade fever of 37.7 °C. In view of his clinical decline and limited tolerance to further chemotherapy, he was referred to a programme of home palliative care.

Within 24 h, he required admission to a local hospital nearby his residence due to a fever of up to 38 °C. Empirical therapy with meropenem 2 g every 8 h was started, but given his frailty and underlying disease, curative management was not pursued. Shortly afterwards, he was transferred to a medium-stay palliative care unit, where therapeutic efforts were restricted to comfort measures. The patient passed away within 24 h of admission to this facility.

## 3. Discussion

*Rhodotorula mucilaginosa* is a pigmented yeast classified as a basidiomycetous fungus. It belongs to the family Sporidiobolaceae within the order Sporidiobolales [1]. *Rhodotorula* spp. are widely distributed in the environment, being isolated from soil, water, and air, as well as from human skin and hospital surfaces [2]. Their ability to colonise indwelling medical devices is enhanced by biofilm formation, which also contributes to antifungal resistance [3].

For a long time, *Rhodotorula* spp. was considered mainly an environmental contaminant or a harmless coloniser with little clinical significance. However, recent reports have shown that isolating this yeast from normally sterile sites should not be ignored. In clinical practice, *Rhodotorula* spp. has been associated with persistent bloodstream infections and clinically significant disease. This is particularly true among patients with haematological malignancies, prolonged neutropenia, or indwelling central venous catheters, where targeted antifungal treatment is often required. Several risk factors for fungemia have been identified, including severe immunosuppression, parenteral nutrition, and extended use of broad-spectrum antibiotics [2,4]. The true incidence of *Rhodotorula* fungemia is likely underestimated, as misidentification and underreporting remain common in routine practice. Mortality rates associated with *Rhodotorula* spp. fungemia vary from 10% to over 25% even with appropriate antifungal therapy, underscoring the significant clinical impact of this infection. Outcomes are heavily influenced by the degree of underlying immune compromise and the promptness of treatment [5].

This case is particularly noteworthy because it involves fungemia caused by *Rhodotorula mucilaginosa*, a rare opportunistic pathogen. Although fungemia caused by this yeast is already uncommon, an abdominal origin is especially unusual, as most reported cases involve central venous catheters. To our knowledge, only three other cases of *Rhodotorula mucilaginosa* fungemia with suspected intestinal origin have been published [6,7,8]. Rajmane et al. [6] described a middle-aged patient with peritonitis and septic shock following duodenal perforation. Hirano et al. [7] reported a patient with rheumatoid arthritis on long-term low-dose corticosteroids and sulfasalazine, in whom intestinal translocation was suspected but not definitively confirmed. Spiliopoulou et al. [8] presented a woman with ovarian cancer who underwent multiple gastrointestinal surgeries and subsequently developed postoperative fungemia.

Comparing these cases with our own highlights notable differences in host background, abdominal source, catheter use, and outcomes. Rajmane’s [6] patient had no chronic immunosuppression, Hirano’s [7] had mild drug-related immunosuppression, and Spiliopoulou’s [8] had ovarian cancer but no reported chemotherapy. In contrast, our patient was profoundly immunosuppressed due to acute leukaemia and severe neutropenia. A central venous catheter was absent in Hirano’s [7] case, not specified in Rajmane’s [6] and Spiliopoulou’s [8], but was present in ours. The abdominal focus was duodenal perforation in Rajmane’s case [6], suspected intestinal translocation in Hirano’s [7], postoperative intestinal necrosis in Spiliopoulou’s [8], and a contained jejunal perforation in our patient. Amphotericin B was used in all cases. Rajmane’s [6] and Hirano’s [7] patients recovered, while Spiliopoulou’s [8] patient had a fatal outcome. Our patient responded to liposomal amphotericin B but died shortly afterwards, likely due to progression of the underlying haematological disease. Taken together, these findings suggest that *Rhodotorula mucilaginosa* caused a true fungemia, but the patient’s later deterioration was mainly related to haematological progression and concomitant aspergillosis, rather than persistent *Rhodotorula* infection. A summary of the main clinical features of these cases, along with our own, is presented in Table 1 for easier comparison.

Diagnosing *Rhodotorula* spp. infections relies on culture, as common fungal biomarkers are unreliable. Specifically, *Rhodotorula* spp. contain low levels of (1→3)-β-D-glucan (BDG) in their cell wall, which explains the limited usefulness of BDG testing in this context. Its relatively slow growth in culture and absence of reliable biomarkers can delay recognition in laboratories, further complicating early diagnosis and treatment [9]. Accurate identification often requires advanced methods such as mass spectrometry or molecular techniques, which are not routinely available in many clinical laboratories. In this case, the isolate was identified by mass spectrometry (MALDI-TOF, Bruker, Bremen, Germany) with a very high confidence score according to international libraries of ribosomal protein spectra [10]. For *Rhodotorula* spp., there are no clinical breakpoints or epidemiological cut-off values (ECOFFS), neither from the Clinical and Laboratory Standards Institute (CLSI) nor from the European Committee on Antimicrobial Susceptibility Testing (EUCAST). The choice of antifungal therapy is usually based on MIC distributions published in the literature [10].

Beyond diagnostic and treatment considerations, intrinsic resistance in *Rhodotorula* spp. is well documented. The genus shows consistent resistance to triazoles and echinocandins. In *Rhodotorula* spp., triazoles poorly bind to lanosterol 14α-demethylase, their target enzyme, making the inhibition of ergosterol synthesis ineffective. Moreover, the cell wall and membrane restrict drug penetration, and biofilm formation further enhances resistance [1,11]. As noted earlier, *Rhodotorula* spp. contains only small amounts of (1→3)-β-D-glucan in its cell wall, which limits the utility of the BDG test and explains its natural resistance to echinocandins. These characteristics render both triazoles and echinocandins ineffective, justifying amphotericin B as the treatment of choice, supported by in vitro susceptibility and limited clinical data [1,11,12].

In our patient, antifungal susceptibility testing showed resistance to fluconazole (MIC > 256 µg/mL), itraconazole (MIC 2 µg/mL), voriconazole (MIC 8 µg/mL), posaconazole (MIC 2 µg/mL), isavuconazole (MIC 1 µg/mL), and all echinocandins (MICs > 8 µg/mL). Conversely, amphotericin B had a low MIC (1 µg/mL), aligning with published in vitro data and limited clinical experience, supporting its role as the most reliable therapy for *Rhodotorula mucilaginosa*. Based on these findings, the isolate was interpreted as susceptible to amphotericin B, which was therefore selected as the treatment in this case [1,11]. Susceptibility testing was performed using Sensititre YeastOne (Thermo Fisher Scientific, Basingstoke, UK), a commercial method conducted according to CLSI guidelines.

We considered adding 5-flucytosine, but ruled it out due to potential toxicity and lack of an intravenous formulation. Fosmanogepix was also evaluated under compassionate use, given limited treatment options. This decision was supported by numerous in vitro studies showing activity against *Rhodotorula mucilaginosa*. Notably, manogepix—the active form of the drug—showed low MICs despite the intrinsic resistance of *Rhodotorula* spp. to azoles and echinocandins [13,14]. Recent reviews have identified fosmanogepix as a promising agent for treating rare fungal infections, especially in complex or refractory cases [15]. However, it was ultimately not used, as the patient showed a favourable clinical response to high-dose liposomal amphotericin B (5 mg/kg/day).

The infection developed while the patient was receiving antifungal prophylaxis with fluconazole, which may have contributed to selecting this intrinsically resistant organism. Additionally, the patient experienced a breakthrough fungal infection while on echinocandin therapy, highlighting the difficulties in managing fungal infections in severely immunocompromised hosts.

Recently, posaconazole has been recommended as the preferred prophylactic agent in patients with acute leukaemia and prolonged neutropenia, as clinical trials have shown superior protection against invasive fungal disease compared with fluconazole [16]. This benefit mainly stems from its reliable activity against filamentous fungi, especially *Aspergillus* spp., which remain the most frequent cause of breakthrough infection in this group. Still, fluconazole remains widely used in clinical practice due to its good safety profile and broad availability, particularly in resource-limited settings where access to newer antifungals may be restricted [16,17]. In our case, switching to posaconazole might have reduced the risk of invasive aspergillosis, but it would not have prevented the current *Rhodotorula mucilaginosa* fungemia, as this yeast is inherently resistant to both fluconazole and posaconazole. This underscores that even with optimal prophylaxis, rare yeasts outside the triazole spectrum can still emerge in severely immunocompromised patients.

Coinfections with yeasts and moulds are rare but possible in severely immunosuppressed patients [18,19]. The development of a mould infection while on liposomal amphotericin B suggests that the patient was already infected upon admission. Given that he presented with cellulitis upon arrival, the cutaneous lesion likely represented an early sign of invasive *Aspergillus fumigatus* infection. Therefore, the subsequent positive cultures from the arm may indicate residual microbial presence from a pre-existing infection that was already being effectively treated at the time of sampling. Considering the isolation of *Aspergillus* from the skin ulcer and the increased 18F-FDG uptake in the left upper lobe on PET-CT, the infection was managed as a probable invasive fungal disease despite negative serum galactomannan and BDG results. A few weeks later, the diagnosis was confirmed through skin biopsy. Further outpatient follow-up by the Infectious Diseases team was considered. However, it could not be carried out due to the patient’s worsening haematological condition and eventual shift to palliative care.

This case report has several limitations. First, the abdominal origin of the fungemia could not be conclusively established, as no direct microbiological evidence was obtained from the jejunal lesion. Second, molecular studies of resistance mechanisms were not performed. While such analyses could have provided additional insights, the antifungal profile of our isolate was consistent with *Rhodotorula*’s known resistance patterns. Third, the co-occurrence of *Aspergillus fumigatus* infection complicates the assessment of *Rhodotorula*’s exact contribution. Fourth, follow-up was limited by the patient’s rapid decline and transition to palliative care, preventing long-term treatment evaluation. Finally, as with any single case report, the findings should be interpreted with caution and not generalised until confirmed by systematic studies.

To our knowledge, this is one of the few documented cases of *Rhodotorula mucilaginosa* fungemia with an apparent abdominal source. It also appears to be among the first reported cases in the context of fluconazole prophylaxis, which may have selected for this resistant intestinal yeast and possibly contributed to the development of *Aspergillus fumigatus* infection.

## 4. Conclusions

In summary, this case highlights the difficulties in managing rare fungal infections in haematological patients. Severe immunosuppression and prolonged neutropenia increase the risk of unusual pathogens that are difficult to diagnose and treat. It also underscores the limitations of fluconazole prophylaxis, which lacks activity against moulds and yeasts such as *Rhodotorula* spp. Greater awareness, systematic reporting, and timely adjustments to therapy are essential in high-risk settings to improve patient outcomes, as timely clinical decisions can significantly influence prognosis, therapeutic success, and overall survival.

## Figures and Tables

**Figure 1 jof-11-00723-f001:**
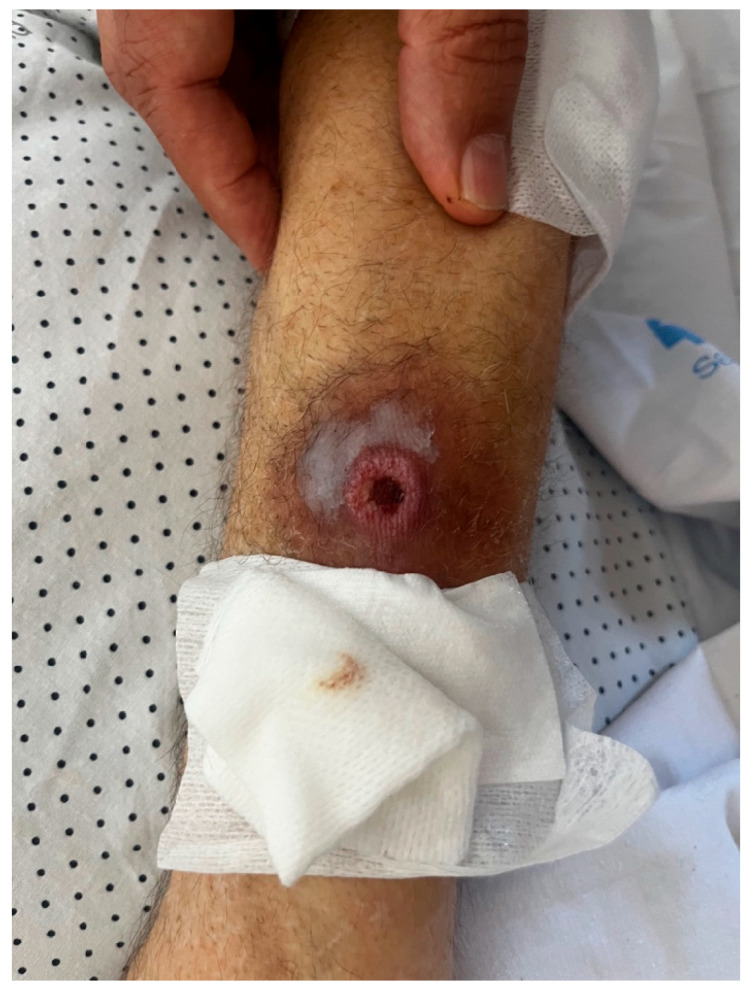
Ulcerative lesion on the patient’s right forearm.

**Table 1 jof-11-00723-t001:** Clinical characteristics of published cases of *Rhodotorula mucilaginosa* fungemia with suspected abdominal origin, compared with the present case.

Author	Immunosuppression	Central Venous Catheter	Abdominal Source	Antifungal Therapy	Outcome
Rajmane et al. [3]	No immunosuppression	Not specified	Duodenal perforation	Amphotericin B (formulation not specified)	Recovery
Hirano et al. [4]	Rheumatoid arthritis; prednisolone 5 mg/day + salazosulfapyridine 1000 mg/day	Absent	Suspected intestinal translocation (not proven)	Liposomal amphotericin B	Recovery
Spiliopoulou et al. [2]	Ovarian cancer; multiple abdominal surgeries, no chemotherapy reported	Not specified	Postoperative intestinal necrosis	Amphotericin B (formulation not specified)	Death
Present case. Fortún et al.	Acute myeloid leukaemia; grade IV neutropenia	Present (PICC)	Contained jejunal micro-perforation	Liposomal amphotericin B	Initial response; death shortly after due to haematological progression

## Data Availability

The original contributions presented in this study are included in the article. Further inquiries can be directed to the corresponding author.

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
