# Peer review of "When Fungal Prophylaxis Fails: A Rare Case of Rhodotorula mucilaginosa Fungemia with Suspected Abdominal Origin"

_jof, 2025, doi:10.3390/jof11100723_

Round 1
Reviewer 1 Report
Comments and Suggestions for Authors
The manuscript titled "When Fungal Prophylaxis Fails: A Rare Case of Rhodotorula mucilaginosa Fungemia with Suspected Abdominal Origin" written by Asensi-Díaz et al. is a very interesting case presentation. Rhodotorula spp. is an opportunistic yeast that rarely causes infections. The study is quite well-documented, and while the specific source of the fungemia remained undetected, I believe that the manuscript could contribute in raising awareness about this pathogen. The authors presented the case in an easy to understand and follow way.
However, I have some suggestions regarding the manuscript:
- in the abstract, none of the microorganism names are written in italic (e.g., Rhodotorulla - line 9, 13, 15; Klebsiella oxytoca, Aspergillus fumigatus - line 12, etc.). The same thing happens everywhere in the text. Please also check through the text and correct everywhere
- the manuscript completely lacks the introduction section
- lines 60-62 - I think it would be worth mentioning the MICs for the resistant antifungals as well, as in the discussion section the values from another article are mentioned and could be compared
- in table 1, several other articles showing case presentations are compared. I think the last line from the table is the current study, based on the data, but this is not very clear because the authors are never mentioned in the manuscript and in the system the first author is someone else? I think it would be best if you just changed it to Our article/Current case presentation or something similar
Author Response
- the manuscript completely lacks the introduction section
INTRODUCTION (lines 42-50)
Rhodotorula species are basidiomycetous yeasts commonly found in the environment and classified within the Pucciniomycotina subphylum [1]. For many years, they were considered harmless colonisers or contaminants, but clinical evidence has demonstrated that they can act as opportunistic pathogens, particularly in immunocompromised hosts [2–4]. The most frequent presentation is fungemia, although cases of peritonitis, meningitis, and endocarditis have also been documented [4]. We report a rare case of Rhodotorula mucilaginosa fungemia with a suspected abdominal origin in a neutropenic patient.
- lines 60-62 - I think it would be worth mentioning the MICs for the resistant antifungals as well, as in the discussion section the values from another article are mentioned and could be compared.
Lines 227-229.
In our patient, antifungal susceptibility testing showed resistance to fluconazole (MIC >256 µg/mL), itraconazole (MIC 2 µg/mL), voriconazole (MIC 8 µg/mL), posaconazole (MIC 2 µg/mL), isavuconazole (MIC 1 µg/mL), and all echinocandins (MICs >8 µg/mL).
- in table 1, several other articles showing case presentations are compared. I think the last line from the table is the current study, based on the data, but this is not very clear because the authors are never mentioned in the manuscript and in the system the first author is someone else? I think it would be best if you just changed it to Our article/Current case presentation or something similar.
Table 1. We have added and highlighted in yellow the text “Present Case” in the first cell of the row describing our case
Reviewer 2 Report
Comments and Suggestions for Authors
In this manuscript, the authors describe an interesting case of Rhodotorula mucilaginosa infection in an immunocompromised patient with AML. The case is well presented, but I have a few comments:
- All microbial names must be in italic.
- Was the skin biopsy performed only once or several times? Was Aspergillus fumigatus isolated in all samples?
- Were the skin biopsies also submitted for histopathological examination? If not, please briefly explain why and clarify that without such data it is very difficult to rule out contamination with A. fumigatus.
- Which method was used to determine the MIC and according to which standard was it interpreted (EUCAST or CLSI)?
- In addition to aspergillosis, please comment on other possible causes of increased 18F-FDG uptake in the lungs. It would also be helpful to explain in more detail why bronchoscopy could not be performed and to state more clearly that invasive aspergillosis was very unlikely based on the lab results.
- When referring to reliable identification of this yeast (line 165), please also mention MALDI-TOF MS and provide appropriate references.
- Are ECOFFs available via EUCAST or CLSI? Please describe in more detail how you interpreted the MIC results.
- Some therapeutic decisions are described without adequate explanation. For example:
- Why was vancomycin administered (due to suspected catheter-related infection, cellulitis, or MRSA risk)?
- Why was amphotericin B discontinued despite persistent fever?
- Why was isavuconazole chosen instead of voriconazole?
Stating the reasons for these decisions would make the report more informative.
Author Response
- Was the skin biopsy performed only once or several times? Was Aspergillus fumigatusisolated in all samples? Were the skin biopsies also submitted for histopathological examination? If not, please briefly explain why and clarify that without such data it is very difficult to rule out contamination with fumigatus.
Lines 111-112
This was the only biopsy, and histopathological analysis confirmed true infection by both pathogens, ruling out contamination.
- Which method was used to determine the MIC and according to which standard was it interpreted (EUCAST or CLSI)?
Lines 234-235
Susceptibility testing was performed using Sensititre YeastOne (Thermo Fisher Scientific, UK), a commercial method conducted according to CLSI guidelines.
- In addition to aspergillosis, please comment on other possible causes of increased 18F-FDG uptake in the lungs. It would also be helpful to explain in more detail why bronchoscopy could not be performed and to state more clearly that invasive aspergillosis was very unlikely based on the lab results.
Lines 124-133
Despite ongoing treatment, the patient continued to have a fever, so a Positron Emission Tomography–Computed Tomography (PET-CT) showed increased 18F-fluorodeoxyglucose (18F-FDG) uptake in the left upper lobe of the lung, in the right arm (at the site of the skin lesion), and improvement in the jejunum. This uptake in the lung is not specific and can also be seen with bacterial infection, inflammation, or malignancy. Bronchoscopy was considered as the next diagnostic step, but it was not performed because the patient had severe thrombocytopenia and was very frail, which made the procedure unsafe. Although galactomannan and beta-D-glucan tests were negative, the PET-CT findings, together with the skin biopsy positive for Aspergillus fumigatus, led us to treat the case as probable invasive aspergillosis.
- When referring to reliable identification of this yeast (line 165), please also mention MALDI-TOF MS and provide appropriate references.
Lines 208-211. Reference 11.
Accurate identification often requires advanced methods such as mass spectrometry or molecular techniques, which are not routinely available in many clinical laboratories. In this case, the isolate was identified by mass spectrometry (MALDI-TOF, Bruker) with a very high confidence score according to international libraries of ribosomal protein spectra [11].
- Are ECOFFs available via EUCAST or CLSI?
Lines 212-216.
For Rhodotorula spp., there are no clinical breakpoints or epidemiological cut-off values (ECOFFS), neither from the Clinical and Laboratory Standards Institute (CLSI) nor from the European Committee on Antimicrobial Susceptibility Testing (EUCAST). The choice of antifungal therapy is usually based on MIC distributions published in the literature[12].
- Please describe in more detail how you interpreted the MIC results.
Lines 232- 234.
Based on these findings, the isolate was interpreted as susceptible to amphotericin B, which was therefore selected as the treatment in this case [1,13].
- Some therapeutic decisions are described without adequate explanation. For example:
Why was vancomycin administered (due to suspected catheter-related infection, cellulitis, or MRSA risk)?
Lines 71-74
Vancomycin (1 g every 12 hours) was initiated at this stage to cover Gram-positive pathogens, including MRSA, given the cellulitis and the patient’s risk factors related to severe immunosuppression and frequent hospital exposure.
Why was amphotericin B discontinued despite persistent fever?
Lines 118-122
Vancomycin had been discontinued after 10 days of treatment, and liposomal amphotericin B after 14 days from blood culture clearance, once the patient was in good general condition. The persistent fever was considered more likely related to his underlying haematological disease, which had not yet been treated due to the intercurrent infection.
Why was isavuconazole chosen instead of voriconazole?
Lines 115-118
Isavuconazole was chosen over voriconazole because it offered equivalent efficacy with a more favourable safety profile, fewer drug interactions, and simpler management in this immunocompromised patient.
Round 2
Reviewer 1 Report
Comments and Suggestions for Authors
I would like to thank the authors for taking my suggestions into consideration. The article has been improved and can be accepted in current form.
Author Response
Thank you for the valuable comments.